# Long-term prognosis after acute kidney injury (AKI): what is the role of baseline kidney function and recovery? A systematic review

Simon Sawhney,[1,2] Mhairi Mitchell,[1] Angharad Marks,[1,2] Nick Fluck,[2] Corrinda Black[1]

▶ Prepublication history and additional material is available. To view please visit the journal (http://dx.doi.org/ 10.1136/bmjopen-2014-006497).

[1]Division of Applied Health Sciences, University of Aberdeen, Aberdeen, UK
[2]Renal Unit, Ward 108 Aberdeen Royal Infirmary, Aberdeen, UK

**Correspondence to**
Dr Simon Sawhney;
simon.sawhney@abdn.ac.uk

## ABSTRACT

**Objectives:** To summarise the evidence from studies of acute kidney injury (AKI) with regard to the effect of pre-AKI renal function and post-AKI renal function recovery on long-term mortality and renal outcomes, and to assess whether these factors should be taken into account in future prognostic studies.

**Design/Setting:** A systematic review of observational studies listed in Medline and EMBASE from 1990 to October 2012.

**Participants:** All AKI studies in adults with data on baseline kidney function to identify AKI; with outcomes either stratified by pre-AKI and/or post-AKI kidney function, or described by the timing of the outcomes.

**Outcomes:** Long-term mortality and worsening chronic kidney disease (CKD).

**Results:** Of 7385 citations, few studies met inclusion criteria, reported baseline kidney function and stratified by pre-AKI or post-AKI function. For mortality outcomes, three studies compared patients by pre-AKI renal function and six by post-AKI function. For CKD outcomes, two studies compared patients by pre-AKI function and two by post-AKI function. The presence of CKD pre-AKI (compared with AKI alone) was associated with doubling of mortality and a fourfold to fivefold increase in CKD outcomes. Non-recovery of kidney function was associated with greater mortality and CKD outcomes in some studies, but findings were inconsistent varying with study design. Two studies also reported that risk of poor outcome reduced over time post-AKI. Meta-analysis was precluded by variations in definitions for AKI, CKD and recovery.

**Conclusions:** The long-term prognosis after AKI varies depending on cause and clinical setting, but it may also, in part, be explained by underlying pre-AKI and post-AKI renal function rather than the AKI episode itself. While carefully considered in clinical practice, few studies address these factors and with inconsistent study design. Future AKI studies should report pre-AKI and post-AKI function consistently as additional factors that may modify AKI prognosis.

**Strengths and limitations of this study**

- This systematic review followed strict inclusion, exclusion and quality assessment criteria to summarise the available evidence regarding the role of pre-acute kidney injury (AKI) baseline and post-AKI recovery of renal function in long-term AKI outcomes.
- Few studies reported long-term AKI outcomes stratified by pre-AKI and post-AKI factors. Quantitative meta-analysis was precluded by heterogeneity in design of included studies.
- The review includes papers from Medline and EMBASE up to October 2012. It may potentially have missed studies published in 2013–2014 or available in other databases.

## INTRODUCTION

Acute kidney injury (AKI) affects an estimated 13–18% of hospitalised patients,[1] frequently under the care of specialties other than nephrology. With the advent of an internationally agreed definition based on changes in serum creatinine and urine output,[2] there is now increasing awareness of the poor outcomes suffered by such patients and this has been accompanied by an emphasis on early detection in an effort to improve patient safety and outcomes.[1] Some of the poor outcomes (including increased mortality[2 3] and development of chronic kidney disease (CKD)[4–6]) are increasingly described after hospital discharge, but this is variable and factors associated with long-term prognosis are poorly understood.[1] Recent guidelines from the National Institute for Health and Care Excellence (NICE) call for more studies into AKI, with non-AKI comparators, but without specifying which factors should be included in study design.

AKI occurs in many different situations and the context is likely to be important when studying onset of AKI and assessing future outcome risk. In particular, clear knowledge of prior kidney (*pre-AKI* baseline) function is essential to distinguish AKI from CKD. This 'baseline function' is typically a creatinine measurement prior to hospitalisation, but may not be available. This can be solved in clinical practice using good clinical judgement, but in epidemiological studies patients with missing baseline values are either assumed to be normal, estimated from other results or excluded. Baseline function is required for grading AKI severity; as a reference for establishing if recovery is complete; and as a means of stratifying patients with and without *pre-AKI* CKD. It is intuitive that AKI in patients with advanced baseline *pre-AKI* CKD could present and behave differently,[7] thus studies without baseline function are at risk of selection bias and misclassification of CKD as AKI.

After an episode of AKI (for instance at hospital discharge or clinic review), the physician also has an opportunity to consider a patient's most recent *post-AKI* kidney function. Accounting for *post-AKI* recovery from AKI (eg, return of creatinine to within 20% pre-AKI creatinine[2]) may assist future risk assessment. However, the timing of this *post-AKI* assessment may also influence its predictive ability, since clinical course may vary, with recovery and deterioration possible and not necessarily at a constant rate.[8]

Thus, while there is evidence that AKI may have a poor overall prognosis, in this systematic review we seek to evaluate whether previous observational studies have demonstrated the modifying contribution of *pre-AKI* baseline function, *post-AKI* recovery and the timing of outcomes. We also assess whether stratification by these factors should be necessary in future prognostic studies and might provide valuable information to the physician making a risk assessment.

## METHODS

A systematic review of observational studies was undertaken in accordance with guidelines for meta-analysis of observational studies in epidemiology (MOOSE).[9] Medline and EMBASE were searched from January 1990 through October 2012 using Medline subject heading (MeSH) terms and free text for AKI (acute renal failure, acute kidney disease, acute kidney injury, acute dialysis, RIFLE, AKIN) and prognosis (prognosis, survival, mortality, follow up, progression, chronic kidney disease,

renal replacement therapy, chronic dialysis, end-stage kidney disease; see online supplementary material). Reference lists of relevant studies and review articles were also searched. There were no language restrictions.

Studies were included if AKI was defined with: creatinine changes within a specified time interval, hospital episode coding for AKI, or the initiation of acute renal replacement therapy (RRT). Studies had to report either mortality or a CKD outcome (development of CKD, progression of CKD severity, or end-stage kidney disease (ESKD)). Studies were included that reported adults (age over 18 years), had at least 50 participants with AKI surviving to hospital discharge, and had follow-up of at least 1 year. Studies of those with only specialised conditions (eg, cancer, chemotherapy, transplantation) were excluded.

Two reviewers independently screened titles, abstracts and full papers against the inclusion and exclusion criteria. Where disagreement could not be resolved by discussion, a third author was available, but was not required. One researcher extracted information from the included studies into a data extraction proforma, with confirmation by a second reviewer.

### Quality assessment

Six quality criteria (based on guidelines for assessing the quality of prognostic studies[10]) with a specific focus on the key areas of potential bias and heterogeneity in studies identifying AKI and subsequent outcomes, were used to assess the included studies (figure 1).

### Analysis

Study characteristics and quality assessment information were tabulated and described. Multiple papers from the same study data set were presented together with findings reported from the most complete paper.

Quality criteria 1–3 limited analysis to studies that accurately identified patients with AKI and restricted to those who survived the acute event, without significant potential for misclassification of patients with CKD. If criteria 1–3 were not met, the study was excluded from further analysis.

Further analysis was limited to studies that analysed patients separately according to one of quality criteria 4–6 (analysis by pre-AKI baseline, post-AKI recovery, or timing of outcomes).

The remaining studies, satisfying criteria 1–3 and one of 4–6, were then summarised in two tables for mortality

**Figure 1** Quality assessment criteria (AKI, acute kidney injury; CKD, chronic kidney disease; eGFR, estimated glomerular filtration rate).

| 1 | Was there a clear objective description for AKI and comparators? |
|---|---|
| 2 | Was there a pre AKI assessment of kidney function (creatinine or CKD code) to identify AKI and baseline kidney function? |
| 3 | Did the study focus specifically on AKI survivors at hospital discharge or at least 30 days for calculating outcomes? |
| 4 | Was pre-AKI baseline CKD (eGFR <60ml/min/1.73m$^2$) used to stratify or analyze outcomes separately against a non-AKI comparator? |
| 5 | Was post-AKI renal recovery clearly described and used to stratify or analyses outcomes separately against a non-AKI comparator? |
| 6 | Was the timing of risk prediction and outcomes considered as a potential source of variation in findings? |

and CKD outcomes. Clinical setting of AKI was described for each table as follows: intensive care (intensive therapy unit (ITU): patients admitted to ITU, regardless of RRT use), postoperative, cardiac (patients presenting following a myocardial infarction or coronary angiogram), unselected (studies of hospital admissions where the clinical settings above did not describe all patients). Retrospective or prospective design was also reported.

Mortality and CKD outcomes were presented separately, reporting study characteristics, population, exposure, comparators, follow-up and measure of outcome. Multivariate adjusted risk ratios for survival and CKD outcomes were displayed graphically if available. If the timing of outcome was considered in a study this was indicated and described.

## RESULTS

The literature search identified 7385 citations. Following review of titles, abstracts and full texts (figure 2),

68 papers describing 61 unique studies were identified (further details summarised in online supplementary material). Study size varied, ranging from 61 to 82 711 individuals with follow-up from 12 to 142 months. Thirty of 61 studies satisfied quality criteria 1–3, reported either mortality or CKD outcomes (and these studies are summarised in table 1).[11–39] Notably, 28/61 studies had insufficient pre-AKI data to avoid misclassification of CKD as AKI. Of 30 remaining studies stratification or separation by exclusion of pre-AKI or post-AKI function subgroups was performed in 16/30 mortality studies with 14/30 not conducting separate analyses. It was also performed in 5/30 CKD outcome studies with 25/30 not conducting separate analyses. As noted in table 2, most studies satisfied quality criteria 1–3 by excluding patient groups rather than by stratifying by pre-AKI or post-AKI function. Only three mortality studies stratified by pre-AKI function and six by post-AKI function. Two CKD studies stratified by pre-AKI and two post-AKI function. Only two mortality studies[8 19] and one CKD

**Figure 2** Study selection and quality assessment (AKI, acute kidney injury; CKD, chronic kidney disease).

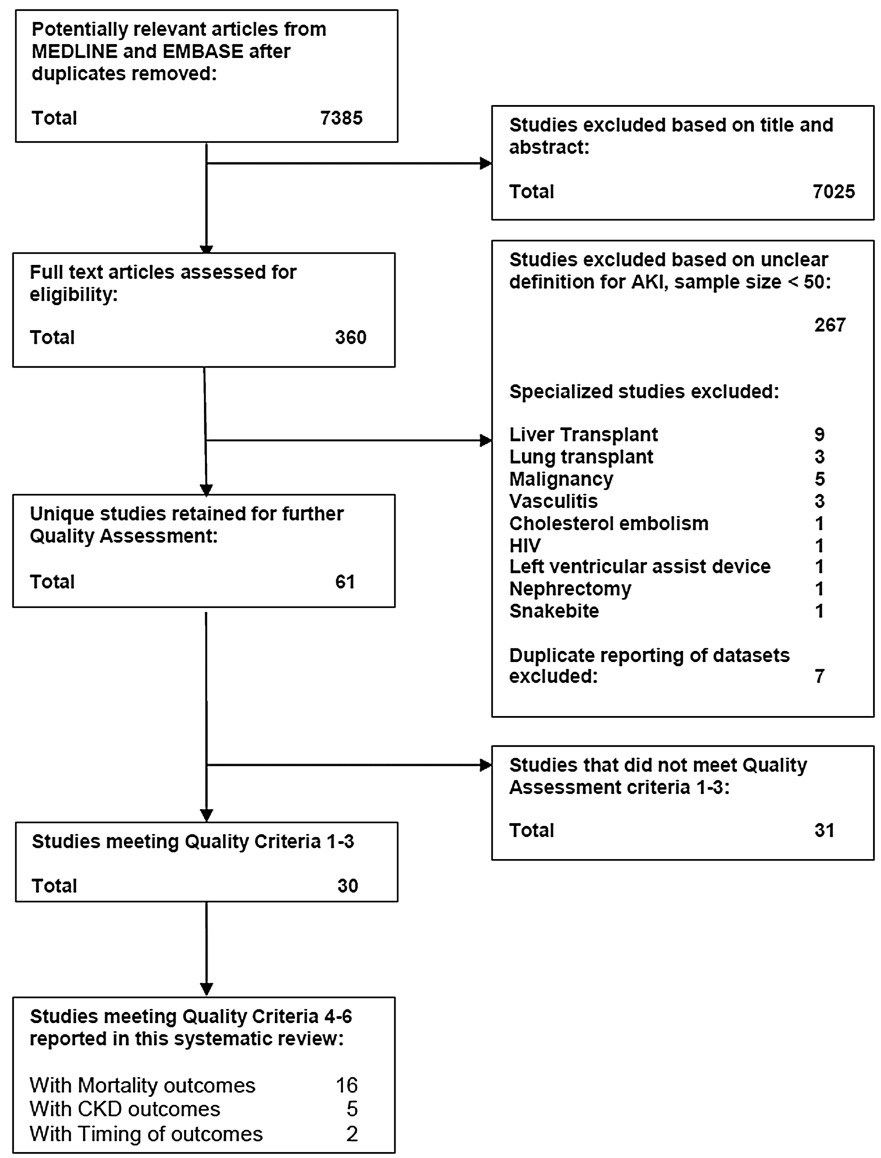

**Table 1** Description and application of quality criteria 4–6 in studies satisfying criteria 1–3

| Paper | N | Clinical setting | Mortality outcomes reported | | CKD outcomes reported | | For either outcome |
|---|---|---|---|---|---|---|---|
| | | | By pre-AKI baseline | By post-AKI recovery | By pre-AKI function | By post-AKI recovery | Timing considered |
| Hoste et al[27] | 82 | ITU | N | N | – | – | – |
| Lopes et al[28] | 61 | ITU | N | N | – | – | – |
| Manns et al[29] | 66 | ITU | N | N | – | – | – |
| Schiffl and Fischer[11] | 226 | ITU | CKD excluded | Y | – | – | – |
| Triverio et al[17] | 95 | ITU | Y | Y | – | – | – |
| Bucaloiu et al[12] | 1610 | Unselected | CKD excluded | Recovered only | CKD excluded | Recovered only | – |
| Hsu et al[13] | 782 | Unselected | CKD only | N | N | N | – |
| Ishani et al[18] | 7197 | Unselected | Y by code alone | N | Y by code alone | N | – |
| Jones et al[14] | 719 | Unselected | CKD excluded | Recovered only | CKD excluded | Recovered only | – |
| Lafrance and Miller[19] | 82 711 | Unselected | Y | Y | – | – | Sensitivity analysis excluding first 6 months |
| Lo et al[30] | 343 | Unselected | N (excluded only if eGFR <45) | N | – | – | – |
| Ng et al[31] | 262 | Unselected | N | N | – | – | – |
| Ponte et al[21] | 177 | Unselected | N (Cr >1.4 excluded) | Y | – | – | – |
| Wald et al[32] | 41 327 | Unselected | N | N | N | N | – |
| Gupta et al[33] | 143 | Cardiac | N | N | – | – | – |
| Kimura et al[34] | 81 | Cardiac | N | N | – | – | – |
| Lindsay et al[35] | 179 | Cardiac | Y | N | – | – | – |
| Maioli et al[15] | 167 | Cardiac | CKD only | Y | – | – | – |
| Rihal et al[36] | 185 | Cardiac | N | N | – | – | – |
| Roghi et al[37] | 106 | Cardiac | N | N | – | – | – |
| Brown et al[22] | 1886 | Postoperative | N | Y | – | – | – |
| Coca et al[23] | 6257 | Postoperative | N | Y | – | – | – |
| Ishani et al[8] | 4053* | Postoperative | N | N | N | N | Changing HRs for mortality and CKD outcomes in time intervals |
| Kheterpal et al[16] | 101 | Postoperative | CKD excluded | N | – | – | – |
| Loef et al[24] | 145 | Postoperative | N | Y | – | – | – |
| Luckraz et al[38] | 53 | Postoperative | N | N | – | – | – |
| Mehta et al[25] | 2083 | Postoperative | N (excluded only Cr>2) | Y | – | – | – |
| Swaminathan et al[39] | 1113 | Postoperative | N | N† | – | – | – |
| Van Kuijk et al[26] | 120 | Postoperative | N | N | CKD excluded | Y | – |
| Wu et al[20] | 4393 | Postoperative | Y by GFR>/<45 | Y | Y by GFR>/<45 | Y | – |

*19 779 in total, 4053 of which had >50% cr rise.
†Outcomes by rate of fall of cr over a 24 h rather than a recovery variable.
–, Outcome not addressed by study; AKI, acute kidney injury; CKD, chronic kidney disease; Cr, creatinine; eGFR, estimated glomerular filtration rate; ITU, intensive therapy unit;; N, outcome addressed but not per quality criterion.

**Table 2** Studies of mortality outcomes in AKI by pre-AKI and post-AKI kidney function

| Mortality | N | Clinical setting | Design | AKI exposure | Comparator | Follow-up | Pre-AKI baseline separation | Post-AKI recovery separation | Recovery definition | Findings |
|---|---|---|---|---|---|---|---|---|---|---|
| Schiffl and Fischer[11] | 226 | ITU | Cohort | Acute RRT | None | 5 years | CKD excluded | Y | Within 10% baseline | Mortality 83% without recovery, 33% with recovery. HR without vs with recovery 4.1† (no non-AKI comparator) |
| Triverio et al[17] | 95 | ITU | Cohort | Acute RRT | None | 3 years | Y | Y | eGFR>60 | Mortality 50% if baseline CKD, 29% if eGFR <60 at discharge, 18% if no renal impairment before or after AKI |
| Bucaloiu et al[12] | 1610 | Unselected | Cohort | Cr rise >50% | No AKI | 3.3 years* | CKD excluded | Recovered only | Within 10% baseline | HR Mortality 1.48† |
| Hsu et al[13] | 782 | Unselected | Cohort | Acute RRT | CKD with no RRT | 4 years | CKD only (eGFR<45) | N | – | HR composite end point of ESKD or mortality 1.3† |
| Ishani et al[18] | 7197 | Unselected | Cohort | Code | No AKI or CKD code | 2 years | Y by code alone | N | – | HR mortality vs no AKI or CKD AKI and CKD 3.24†, AKI 2.48†, CKD 1.45† |
| Jones et al[14] | 719 | Unselected | Cohort | Code | No AKI | 2.5 years* | CKD excluded | Recovered only | Within 10% baseline | HR Mortality 1.08 |
| Lafrance et al[19] | 82 711 | Unselected | Cohort | Cr rise >50% | No AKI | 2.34 years* | Y | Recovered only in a subanalysis | Within 10% baseline | HR mortality for AKI vs no AKI in 90-day survivors 1.41†. Subgroup of 6-month survivors 1.13†. Subgroup with recovery 1.47† |
| Ponte et al[21] | 177 | Unselected | Cohort | Cr rise from<1.4 mg/dL to >2 | None | 7.2 years* | CKD excluded (Cr >1.4 mg/dL) | Y | Cr<1.4 mg/dL | 10-year mortality 40% with recovery, 57% without recovery |
| Lindsay et al[35] | 179 | Cardiac | Cohort | Cr rise 50% from<1.2 mg/dL | No AKI | 1 year | CKD excluded (cr >1.2 mg/dL) | N | – | 1-year mortality 9.5% AKI, 2.7% no-AKI |
| Maioli et al[15] | 167 | Cardiac | Cohort | Cr rise (0.5 mg/dL by 3 days) | No AKI | 3.8 years* | CKD only (eGFR <60) | Y | Within 25% baseline at 3 months | HR mortality for AKI with recovery1.3†, without recovery 2.3† |
| Brown et al[22] | 1886 | Postoperative | Cohort | Cr rise 0.3 mg/dL or 50% | No AKI | 2.6 years* | N | Y | Number of days AKI definition met | HR mortality for AKI vs no AKI by AKI duration 1–2 days 1.51†, 3–6 days 1.74†, >7 days 3.45†, persistent 5.75† |
| Coca et al[23] | 6257 | Postoperative | Cohort | Cr rise 0.3 mg/dL or 50% | No AKI | 3.8 years* | N | Y | Number of days AKI definition met | HR mortality for AKI vs no AKI by AKI duration <2 days 1.15†, 3–6 days 1.5†, >7 days any duration with RRT 2.10† |
| Kheterpal et al[16] | 101 | Postoperative | Cohort | Cr clearance fall to<50 | No AKI | 1 year | CKD excluded | N | – | 1-year mortality 12% AKI, 9% no AKI |
| Loef et al[24] | 145 | Postoperative | Cohort | Cr rise 25% | No AKI | 100 months | N | Y | "Improved to or below the preoperative level" | HR mortality for AKI 1.63†, AKI with recovery 1.66†, AKI without recovery 1.72†, non-recovery vs recovery 1.22 (not significant) |

Continued

**Table 2** Continued

| Mortality | N | Clinical setting | Design | AKI exposure | Comparator | Follow-up | Pre-AKI baseline separation | Post-AKI recovery separation | Recovery definition | Findings |
|---|---|---|---|---|---|---|---|---|---|---|
| Mehta et al[25] | 2083 | Postoperative | Cohort | Cr rise 50% or 0.7 mg/dL | No AKI | 7 years* | N | Y | AKI definition no longer met at 7 days | HR mortality for AKI with recovery 1.21†, without recovery 1.47† |
| Wu et al[20] | 4393 | Postoperative | Cohort | Cr rise >50% | No AKI and no CKD | 4.76 years* | Y by GFR>/<45 | Y | Within 50% baseline | HR mortality for AKI only 1.94†, CKD only 2.64†, AKI and CKD 3.28† Stratified by recovery AKI and CKD with recovery 3.0†, without recovery 4.59† AKI only with recovery 1.96†, without recovery 2.18† CKD without AKI 2.59† |

*Mean/median.
†Statistically significant p<0.05.
—, Outcome not addressed by study; AKI, acute kidney injury; CKD, chronic kidney disease; Cr, serum creatinine; eGFR, estimated glomerular filtration rate; ESKD, end-stage kidney disease (dialysis >90 days); ITU, intensive therapy unit; N, outcome addressed but not per quality criterion; RRT, renal replacement therapy.

outcome study[8] addressed whether the risk of outcomes changed with increasing time from AKI.

Among these studies, definitions varied: AKI was defined by acute RRT,[13] different thresholds for creatinine rise,[15 23–26] or hospital episode code.[14 18] CKD was defined by hospital codes,[18] low estimated glomerular filtration rate (eGFR) for CKD cut-off ($<45$ mL/min/1.73 m$^2$)[13 20] conventional eGFR for CKD ($<60$ mL/min/1.73 m$^2$),[15] varying thresholds based on serum creatinine[21 35] or hospital episode code.[18] Definitions of renal recovery also varied: with definitions of its timing ranging from 3 days[26] to 3 months;[15] and its extent from within 50% of baseline[20] to within 10% of baseline,[14] or below a pre-AKI value.[24]

### Mortality findings

There were 16 mortality studies (table 2). Follow-up ranged from 1 to 7 years and mortality up to 83% for patients with AKI at 5 years. AKI was associated with increased mortality in all but one study[14] regardless of pre-AKI baseline (HR 1.08 to 4.59; figure 3) or recovery of renal function (HR 1.08 to 5.75; figure 4). Five studies did not report non-AKI comparators[11 17 21] or risk ratios.[16 35]

Of the three studies stratified by pre-AKI baseline function, in two the overall prognosis was worse in patients with AKI with prior CKD with doubling of HRs with respect to a non-AKI non-CKD comparator.[15 20] However, in a third study, where comparators were also stratified by eGFR, the independent mortality risk from AKI diminished with advancing CKD.[19] Six studies compared mortality stratifying either by post-AKI recovery (four studies)[15 20 24 25] or by AKI duration (two studies).[22 23] All but one[15] were postoperative studies with different thresholds for defining recovery. Risk from incomplete recovery varied from minimal with overlapping CIs[15 20 24 25] to greater than double when AKI lasted more than 1 week.[22 23] Only one study combined stratification by pre-AKI baseline and by post-AKI recovery (defining recovery to 50% baseline, CKD as eGFR$<45$ mL/min/1.73 m$^2$) with non-recovery as well as baseline CKD adding to mortality risk.[20]

### CKD outcomes

In five studies, there was marked variation in CKD outcomes following AKI (HR 1.91 to 213) depending on pre-AKI baseline and post-AKI recovery (table 3 and figure 5).[12 14 18 20 26] In the two studies stratifying by pre-AKI function, AKI with baseline CKD was associated with a substantially greater (fourfold to fivefold) risk of ESKD than AKI in cases without prior CKD[18 20] although notably one study defined CKD at eGFR$<45$ mL/min/1.73m$^2$[20] and the other by a hospital code.[18] The two studies stratifying by post-AKI function had different findings. In one postoperative study[26] there was no association between non-recovery post-AKI (vs patients with recovery) and additional CKD progression, when recovery was measured at day three,[26] but association was substantial (5–10-fold) in the other study where recovery was

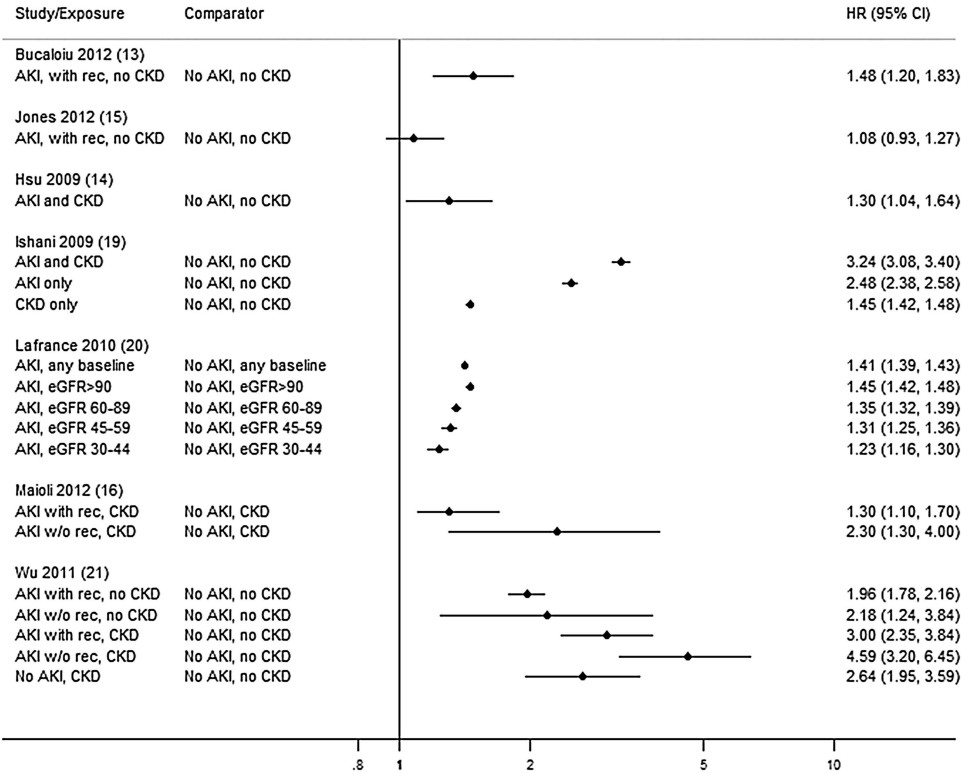

**Figure 3** Mortality—by pre-AKI baseline (AKI, acute kidney injury; CKD, chronic kidney disease; eGFR, estimated glomerular filtration rate; w/o rec, without recovery; with rec, with recovery).

defined at hospital discharge.[20] Again only one study stratified by both baseline and recovery, finding both associated with poorer prognosis.[20]

## Timing of outcomes

Two studies considered whether HRs changed over time following the AKI episode.[8] [19] Lafrance *et al*,[19] performed a sensitivity analysis of their study of AKI survivors and mortality, limiting analysis from 90-day to 6-month AKI survivors. The mortality HR (AKI vs non-AKI) fell from 1.41 (1.39 to 1.43) in 90-day survivors to 1.13 (1.11 to 1.14) in 6-month survivors.[19] Ishani *et al*[8] reported changes in HR of mortality and CKD outcomes from AKI (vs no AKI) when dividing follow-up into 6-month time discrete intervals. For both mortality and CKD outcomes, greatest relative risk (vs non-AKI) was in the first 6 months, with marked attenuation of risk (but still statistically significant) from 1 year.[8] While recognising that risk may not be constant, the study also did not consider the importance of stratifying patients by pre-AKI baseline or post-AKI recovery.

## DISCUSSION

The introduction of a global definition for AKI has led to growing interest in the high incidence and poor outcomes. In this review of 7385 citations, there were only 30 studies that met the initial quality criteria for selecting patients with AKI with sufficient efforts to avoid misclassifying CKD. Of these studies only three mortality

studies stratified by pre-AKI, six by post-AKI function, two CKD outcome studies by pre-AKI and two by post AKI function. Only one study considered both pre-AKI and post-AKI function. Only two studies considered the change in risk over time from AKI.

As with two previous reviews, we note that AKI is associated with overall poorer mortality and CKD outcomes,[3 5] but with substantial variation in the magnitude of risk. We were transparent about the variation in definitions and clinical setting and identified heterogeneity as a problem with the design of AKI studies to date. The two previous reviews included studies with incomplete or estimated data on baseline kidney function,[3 5] but in studies with available data we found that outcomes were modulated by pre-AKI baseline and post-AKI renal recovery. Unfortunately, these studies varied in definitions for AKI, recovery and follow-up preventing pooled meta-analysis. While proteinuria and oliguria may also be important factors, research on their association with long-term prognosis is sparse and we therefore focused on the course of serum creatinine prior and subsequent to the AKI episode.

Pre-AKI baseline CKD was associated with doubling of mortality outcomes and fourfold to fivefold increased risk of CKD outcomes compared to patients with non-AKI non-CKD. Unfortunately, only one study chose CKD comparators for patients with AKI at each level of pre-AKI function, finding that the additional risk from AKI diminished in those already at high risk due to advanced baseline CKD.[19] It is likely the mortality and ESKD risk in advanced CKD is already high and less influenced by AKI.

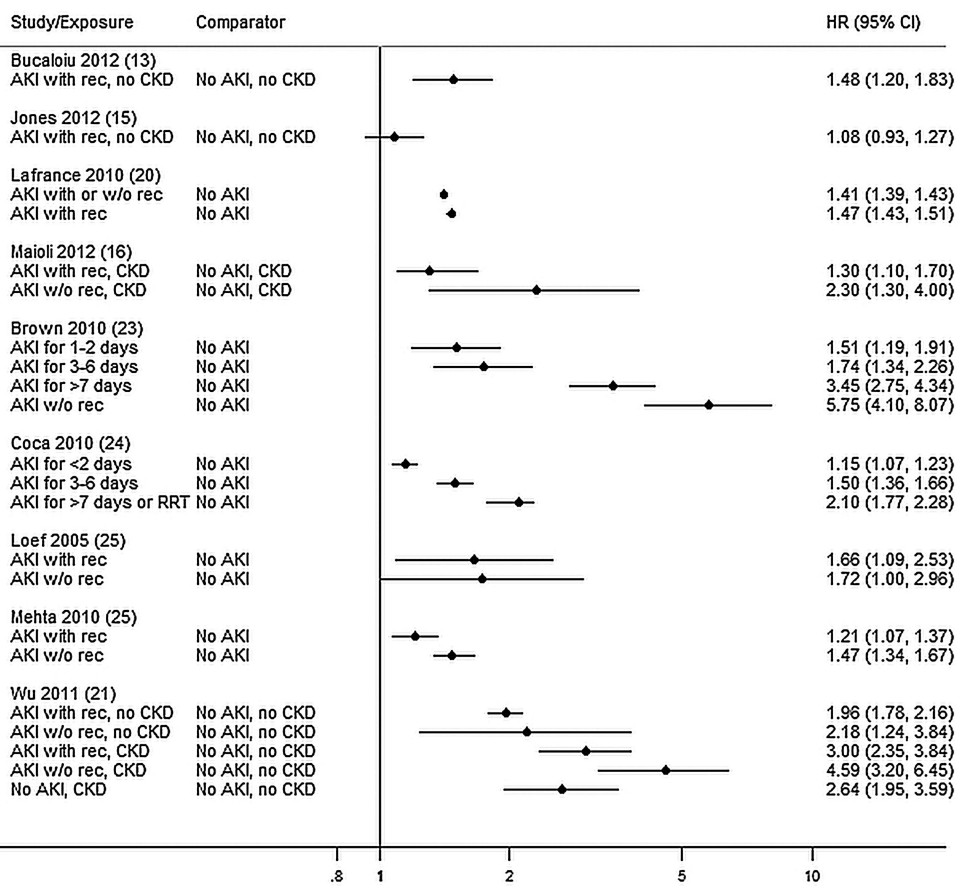

**Figure 4** Mortality—by post AKI recovery (AKI, acute kidney injury; CKD, chronic kidney disease; w/o rec, without recovery; with rec, with recovery).

Non-recovery of kidney function was associated with greater mortality and CKD outcomes in some studies, but the finding was inconsistent and dependent on study design. Studies with a less stringent threshold[20] and later point of assessment[15] showed a poorer prognosis with non-recovery, highlighting the significance of arbitrary thresholds in study design. We also note a further study outside the time period in this review reporting a cohort with poorer outcomes with non-recovery after AKI, in line with our overall findings.[40]

Increased risk of poor outcomes following AKI may be greatest early post-AKI, and therefore the timing of clinical assessment is important. Despite this, most studies followed a 'proportional hazards' assumption (ie, that the relative risk of AKI vs non-AKI comparator was constant over time), but this may not hold. Only two studies considered the timing of assessment post-AKI and found that additional risk of mortality as well as renal progression may diminish over time,[8 19] but neither combined this with adjustment nor stratification by post-AKI recovery. Both factors are required to assess if the rate of CKD progression is more rapid in AKI than non-AKI (but baseline CKD) comparators with equal kidney function at the time of hospital discharge. Thus, understanding of how mortality and renal progression risks change over time remains inadequately addressed.

Overall while our findings broadly agree with previous reviews,[3 5] we found that some of the variation in AKI prognosis may be explained by pre-AKI CKD and post-AKI non-recovery which both lead to a poorer prognosis. This is of great relevance given that these factors were infrequently addressed by stratification, and 28/61 studies had insufficient pre-AKI data to minimise misclassification between CKD and AKI. The relationships are complex with diminishing additional risk with advancing pre-AKI CKD. We also noted a non-linear clinical post-AKI course, with risk prediction dependent on the time point at which risk is assessed, necessitating fixed follow-up points to prevent overestimation of risk in a cohort with shorter follow-up. Importantly, few studies assessed these factors, using different definitions, and with only one study assessing both pre-AKI and post-AKI factors. This review included studies up to October 2012 and in the near future it is likely that AKI registries with additional routine outcome data will become available. Therefore there is now an opportunity to ensure that a concordance exists in definitions and reporting of future AKI studies before this review is revisited.

The NICE guidelines call for further studies of long-term AKI prognosis but without specific detail on which factors should be included. We suggest that future studies should adopt consistent definitions, incorporate

**Table 3** Studies of CKD outcomes in AKI by pre-AKI and post-AKI kidney function

| Progression | N | Clinical setting | Design | AKI exposure | Comparator | Follow-up | Pre-AKI baseline separation | Post-AKI recovery separation | Recovery definition (discharge unless otherwise stated) | Findings |
|---|---|---|---|---|---|---|---|---|---|---|
| Bucaloiu et al[12] | 1610 | Unselected | Cohort | Cr rise >50% | No AKI | 3.3 years* | CKD excluded | Recovered only | Within 10% baseline | Development of new CKD stage 3/1000 person-year 28.1 AKI vs 13.1 no AKI (HR 1.91†) |
| Ishani et al[18] | 7197 | Unselected | Cohort | Code | No AKI or CKD code | 2 years | Y by code alone | N | – | ESKD/1000 person-year (HR vs no code) AKI and CKD 79.45 (HR 41.2†) AKI 24.52 (HR 13†) CKD only 19.88 (HR 8.43†) No Code 2.08 (reference) |
| Jones et al[14] | 719 | Unselected | Cohort | Code | No AKI | 2.5 years* | CKD excluded | Recovered only | Within 10% Baseline | HR for new CKD stage 3 (15% of AKI vs 3% of no AKI patients) 3.82† |
| van Kuijk et al[26] | 493 | Postoperative | Cohort | 10% Cr change by day 2 post-op | No AKI | 5 years* | CKD excluded | Y | Within 10% baseline at day 3 | Development of new CKD 11% if no AKI, 32% if AKI recovered (HR 3.4†), 36% if AKI without recovery (HR3.6†) |
| Wu et al[20] | 4393 | Postoperative | Cohort | Cr rise (RIFLE, or 50% rise if previous CKD) | No AKI and no CKD | 4.76 years* | Y by GFR>/<45 | Y | Within 50% baseline | Development of ESKD (vs no code) AKI only 4.64†, CKD only 40.86†, AKI and CKD 91.69† Stratified by recovery AKI+CKD+without recovery 212.73† AKI+CKD+recovery 74.07† AKI only+without recovery 60.95† AKI only+recovery 4.5† |

*Mean/median.
†Statistically significant p<0.05.
–, Outcome not addressed by study; AKI, acute kidney injury; CKD, chronic kidney disease; Cr, serum creatinine; eGFR, estimated glomerular filtration rate; ESKD, end-stage renal disease (dialysis >90 days); ITU, intensive therapy unit; N, outcome addressed but not per quality criterion; RRT, renal replacement therapy.

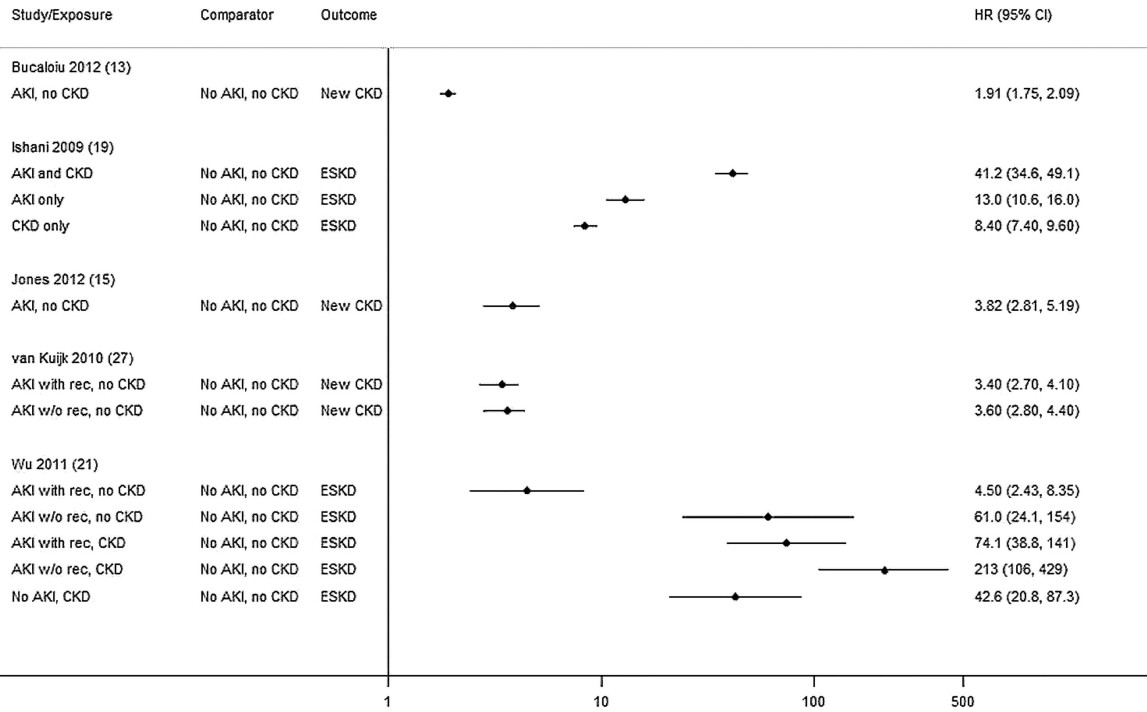

**Figure 5** CKD outcomes (w/o rec, without recovery; with rec, with recovery; ESKD, end-stage kidney disease).

baseline and recovery function and consider the timing of outcomes during follow-up. We therefore suggest that future long-term prognostic studies should adopt the KDIGO definition for identifying and stratifying AKI,[41] the KDIGO CKD stages for stratifying AKI and non-AKI comparator groups by pre-AKI eGFR,[42] the UK Renal Association definition of renal recovery for stratifying recovery (within 20% of baseline)[2] and fixed interim points for establishing recovery (7, 30 and 90 days) and excluding early outcomes (up to 30, 90 and 180 days).

## CONCLUSION

AKI has a poor, but variable prognosis influenced by clinical setting, underlying cause and comorbidity. There is a recognised need to understand which patients are at greater long-term risk and when AKI may carry additional risk beyond the underlying illness. In this review, pre-AKI CKD was associated with doubling of mortality outcomes and a fourfold to fivefold increase in renal outcomes. Non-recovery of kidney function may also affect prognosis, but the magnitude depended on both recovery definition and the timing of assessment. These factors are already considered by physicians in clinical practice, but unfortunately we found that few prognostic studies have addressed them. The heterogeneity of studies and accompanying lack of a clear consistent pattern of effect limit their clinical interpretation. As pre-AKI and post-AKI renal function may influence clinical practice, we suggest that consistent reporting of these factors is needed in future prognostic studies to establish how they modify AKI prognosis. This will inform the clinician as well as future research.

**Contributors** SS, AM, NF and CB contributed to the design of the review, SS and MM acquired the data. SS, AM and CB contributed to analysis. SS wrote the draft and SS, AM, NF and CB critically revised the intellectual content of the work. All authors have given final approval for this paper to be published.

**Funding** SS is supported by a research training fellowship from the Wellcome Trust (Reference Number 102729/Z/13/Z).

**Competing interests** None.

**Provenance and peer review** Not commissioned; externally peer reviewed.

**Data sharing statement** No additional data are available.

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
