## [Reviewer comments · BMJ Open]

Some articles will have been accepted based in part or entirely on reviews undertaken for other BMJ Group journals. These will be reproduced where possible.

ARTICLE DETAILS

TITLE (PROVISIONAL)	Long Term Prognosis after Acute Kidney Injury (AKI) - What is the Role of Baseline Kidney Function and Recovery? A Systematic Review
AUTHORS	Sawhney, Simon; Mitchell, Mhairi; Fluck, Nick; Marks, Angharad; Black, Corrinna

VERSION 1 - REVIEW

REVIEWER	Aled Phillips Institute of Nephrology
REVIEW RETURNED	17-Sep-2014

GENERAL COMMENTS	The main issue with the paper is not the work itself - the idea of a systematic review of the influence of renal function (both pre and post) on AKI outcome is fine. The problem is that the subject area is a messy one in that the "disease" is not a disease but an end result of a heterogeneous set of illnesses and outcome is by the nature of the "disease" influenced by setting, aetiology and co-morbidities, and so pulling out renal function only as a marker of outcome may be somewhat ambitious? This is reflected in the lack of a clear repeatable consistent pattern of effect, such that I'm not entirely sure I know what the take home message is. That is not to say that this is not a worthwhile study but I think that greater emphasis needs to be placed on the other "presumed" factors which influence outcome, which I appreciate may be beyond the scope of this piece of work and a greater recognition that the stated intention of a clean conclusion/association may possibly be beyond what is out in the current literature.
---

REVIEWER	Laurie Tomlinson London School of Hygiene and Tropical Medicine, London, UK I work with this group on some research but have no involvement with this paper whatsoever.
REVIEW RETURNED	29-Sep-2014

GENERAL COMMENTS	This systematic review identifies and examines the evidence for the impact of pre-AKI CKD and degree of post-AKI recovery on long-term outcomes following AKI. It is an excellent piece of work, bringing some important critical thinking to this topic, and highlighting the marked limitations of the evidence. I have only very minor comments which can be left to the discretion of the authors and would recommend publication. Sentences 2 and 3 of the results section in the abstract don't make easy sense until you have read the paper and it would help if they
--

	could be rephrased. In sentence 2 of the introduction, I would think that ‘early detection’ is better than ‘screening’ which implies something more related to chronic conditions. In paragraph 2 of the introduction “This “baseline function” is typically a creatinine measurement prior to hospitalisation, but may not be available or may be estimated if absent.” – I would add an additional sentence about the ‘estimated if absent’ to inform the non-expert reader about the concept of assuming a baseline eGFR and back calculating creatinine, and how this relates primarily to epidemiological studies rather than clinical practice. Discussion page 11/12 line 58/3. It is not clear what the authors mean by “but may be a source of selection bias and misclassification of CKD patients.” This could be more clearly explained. Is the x axis of figure 5 correctly labelled? HRs of 500 seem pretty impressive!
--	--

VERSION 1 – AUTHOR RESPONSE

We would like to thank the editors and reviewers for considering our paper and providing detailed and helpful comments, which have improved the clarity of the paper. We appreciate the opportunity to respond to those comments below.

Editors:	*Please ensure the acronym for NICE is correct (National Institute for Health and Care Excellence)	This has been amended appropriately to “ National Institute for Health and Care Excellence ”
	*Please upload a copy of the MOOSE checklist	This has been performed and is attached
	*Please add as a limitation (or explain why) you only searched 2 databases? In addition, why does the search stop at Oct 2012 – this needs updating to 2014	We searched for observational studies with mortality and renal outcomes for patients with renal disease. Our search strategy therefore included Medline and EMBASE as a focus on European and US medical and clinical literature rather than nursing, allied health or management articles or clinical trials. This replicates the search strategy of the main previous AKI review (Coca et al 2009). We also reviewed all citations to ensure no potentially relevant articles were missed. We have added a limitations statement as suggested. “The review includes papers from Medline and EMBASE up to October 2012. It may potentially have missed studies published in 2013-14 or available in other databases.” Our search strategy was from January 1990 to October 2012. The large number of studies to screen (7385), and resources has meant that it has taken some time to process the data. Our review highlights limitations in the reporting of pre-AKI and post-AKI function in current literature. We may potentially have missed articles published 2013-2014 (and have added this as a limitation), but no new relevant studies were reported at the recent Royal Society of Medicine AKI Frontiers Symposium (September 2014). Of importance our recommendations on future study design are very timely at present when automated AKI detection and AKI registry formation is planned in England, which will lead to the design of new large prognostic studies. An additional search to 2014 would involve substantial additional screening and delay but would be unlikely to find any papers that would change our findings or recommendations. Thus, although we agree that this review should be repeated in the future (we have added two new

		sentences to the discussion reflecting this), we feel an updated review should be after new studies accumulate and we feel that prompt publication of this review now will have a potential to improve the quality and relevance of those new studies. “This review included studies up to October 2012 and in the near future it is likely that AKI registries with additional routine outcome data will become available. Therefore there is now an opportunity to ensure that a concordance exists in definitions and reporting of future AKI studies before this review is revisited.”
Reviewer: 1	The main issue with the paper is not the work itself - the idea of a systematic review of the influence of renal function (both pre and post) on AKI outcome is fine. The problem is that the subject area is a messy one in that the "disease" is not a disease but an end result of a heterogenous set of illnesses and outcome is by the nature of the "disease" influenced by setting, aetiology and co-morbidities, and so pulling out renal function only as a marker of outcome may be somewhat ambitious? This is reflected in the lack of a clear repeatable consistent pattern of effect, such that I'm not entirely sure I know what the take home message is. That is not to say that this is not a worthwhile study but I think that greater emphasis needs to be placed on the other "presumed" factors which influence outcome, which I appreciate may be beyond the scope of this piece of work and a greater recognition that the stated intention of a clean conclusion/association may possibly be beyond what is out in the current literature.	We appreciate these comments from reviewer 1 that AKI is complex. We agree that it is not one single “disease”, but a heterogeneous set of illnesses with the outcome influenced by the clinical context of the event itself in addition to renal specific factors (such as pre and post-AKI function). Nevertheless, such generalisations are made both in clinical practice and research, and it will be necessary for clinicians in England to respond automated AKI “alerts” appraising patients both based on clinical context and changes in kidney function. We also agree that heterogeneity in study designs and clinical settings in the current literature preclude a “clean association” and we have rephrased our abstract and conclusion accordingly. Abstract “The long term prognosis after AKI varies depending on cause and clinical setting, but it may also, in part, be explained by underlying pre-AKI and post-AKI renal function rather than the AKI episode itself. While carefully considered in clinical practice, few studies address these factors and with inconsistent study design. Future AKI studies should report pre-AKI and post-AKI function consistently as additional factors that may modify AKI prognosis.” Conclusion “AKI has a poor, but variable prognosis influenced by clinical setting, underlying cause and comorbidity. There is a recognised need to understand which patients are at greater long-term risk and when AKI may carry additional risk beyond the underlying illness. In this review, pre-AKI CKD was associated with doubling of mortality outcomes and a four-fivefold increase in renal outcomes. Non-recovery of kidney function may also affect prognosis, but the magnitude depended on both recovery definition and the timing of assessment. These factors are already considered by physicians in clinical practice, but unfortunately we found that few prognostic studies have addressed them. The heterogeneity of these studies and accompanying lack of a clear consistent pattern of effect limit their clinical interpretation. As pre and post-AKI renal function may influence clinical practice, we suggest that consistent reporting of these factors is needed in future prognostic studies to establish how they modify AKI prognosis. This will inform both the clinician and future research.”
Reviewer: 2	This systematic review identifies and examines the evidence for the impact of pre-AKI CKD and degree of post-AKI recovery on long-term outcomes following AKI. It is an excellent piece of work, bringing some important critical thinking to this topic, and highlighting the marked limitations of the evidence.	We thank the reviewer for these positive comments.

	I have only very minor comments which can be left to the discretion of the authors and would recommend publication.	
	Sentences 2 and 3 of the results section in the abstract don't make easy sense until you have read the paper and it would help if they could be rephrased.	We have reworded these sentences. "Of 7385 citations, few studies met inclusion criteria, reported baseline kidney function and stratified by pre or post-AKI function. For mortality outcomes, 3 studies compared patients by pre-AKI renal function and 6 by post-AKI function. For CKD outcomes, 2 studies compared patients by pre-AKI function and 2 by post-AKI function."
	In sentence 2 of the introduction, I would think that 'early detection' is better than 'screening' which implies something more related to chronic conditions.	We have changed "screening" to "early detection" in the introduction.
	In paragraph 2 of the introduction "This "baseline function" is typically a creatinine measurement prior to hospitalisation, but may not be available or may be estimated if absent." – I would add an additional sentence about the 'estimated if absent' to inform the non-expert reader about the concept of assuming a baseline eGFR and back calculating creatinine, and how this relates primarily to epidemiological studies rather than clinical practice.	We have added a sentence reflecting the challenges of estimating baseline values in epidemiological studies. "This "baseline function" is typically a creatinine measurement prior to hospitalisation, but may not be available. This can be solved in clinical practice using good clinical judgement, but in epidemiological studies patients with missing baseline values are either assumed to be normal, estimated from other results, or excluded."
	Discussion page 11/12 line 58/3. It is not clear what the authors mean by "but may be a source of selection bias and misclassification of CKD patients." This could be more clearly explained.	We have replaced this sentence with more specific explanation. "...some of the variation in AKI prognosis may be explained by pre-AKI CKD and post-AKI non-recovery which both lead to a poorer prognosis. This is of great relevance given that these factors were infrequently addressed by stratification, and 28/61 studies had insufficient pre-AKI data to minimise misclassification between CKD and AKI."
	Is the x axis of figure 5 correctly labelled? HRs of 500 seem pretty impressive!	We can confirm that figure 5 is correctly labelled. In the study of Wu et al. (2011), the long term risk of end stage kidney disease was substantially higher in patients with pre-AKI CKD and no post-AKI recovery in comparison to patients with no AKI and no pre-AKI CKD, hazard ratio 212.73, 95% confidence interval 105.53-428.83. These hazard ratios are as reported in their paper.

VERSION 2 – REVIEW

REVIEWER	Laurie Tomlinson LSHTM, UK As mentioned, I have been involved in collaborative research with this group but have had no involvement at any stage with this paper.
REVIEW RETURNED	21-Nov-2014

GENERAL COMMENTS	I am satisfied with the amendments the authors have made and am happy to recommend publication
--